# DeepCoder: Learning to Write Programs

**Matej Balog**[*]
Department of Engineering
University of Cambridge

**Alexander L. Gaunt, Marc Brockschmidt,
Sebastian Nowozin, Daniel Tarlow**
Microsoft Research

## Abstract

We develop a first line of attack for solving programming competition-style problems from input-output examples using deep learning. The approach is to train a neural network to predict properties of the program that generated the outputs from the inputs. We use the neural network's predictions to augment search techniques from the programming languages community, including enumerative search and an SMT-based solver. Empirically, we show that our approach leads to an order of magnitude speedup over the strong non-augmented baselines and a Recurrent Neural Network approach, and that we are able to solve problems of difficulty comparable to the simplest problems on programming competition websites.

## 1 Introduction

A dream of artificial intelligence is to build systems that can write computer programs. Recently, there has been much interest in program-like neural network models (Graves et al., 2014; Weston et al., 2015; Kurach et al., 2015; Joulin & Mikolov, 2015; Grefenstette et al., 2015; Sukhbaatar et al., 2015; Neelakantan et al., 2016; Kaiser & Sutskever, 2016; Reed & de Freitas, 2016; Zaremba et al., 2016; Graves et al., 2016), but none of these can *write programs*; that is, they do not generate human-readable source code. Only very recently, Riedel et al. (2016); Bunel et al. (2016); Gaunt et al. (2016) explored the use of gradient descent to induce source code from input-output examples via differentiable interpreters, and Ling et al. (2016) explored the generation of source code from unstructured text descriptions. However, Gaunt et al. (2016) showed that differentiable interpreter-based program induction is inferior to discrete search-based techniques used by the programming languages community. We are then left with the question of how to make progress on program induction using machine learning techniques.

In this work, we propose two main ideas: (1) learn to induce programs; that is, use a corpus of program induction problems to learn strategies that generalize across problems, and (2) integrate neural network architectures with search-based techniques rather than replace them.

In more detail, we can contrast our approach to existing work on differentiable interpreters. In differentiable interpreters, the idea is to define a differentiable mapping from source code and inputs to outputs. After observing inputs and outputs, gradient descent can be used to search for a program that matches the input-output examples. This approach leverages gradient-based optimization, which has proven powerful for training neural networks, but each synthesis problem is still solved independently—solving many synthesis problems does not help to solve the next problem.

We argue that machine learning can provide significant value towards solving Inductive Program Synthesis (IPS) by re-casting the problem as a big data problem. We show that training a neural network on a large number of generated IPS problems to predict cues from the problem description can help a search-based technique. In this work, we focus on predicting an order on the program space and show how to use it to guide search-based techniques that are common in the programming languages community. This approach has three desirable properties: first, we transform a difficult search problem into a supervised learning problem; second, we soften the effect of failures of the neural network by searching over program space rather than relying on a single prediction; and third, the neural network's predictions are used to guide existing program synthesis systems, allowing us to use and improve on the best solvers from the programming languages community. Empirically, we

---

[*]Also affiliated with Max-Planck Institute for Intelligent Systems, Tübingen, Germany. Work done while author was an intern at Microsoft Research.

show orders-of-magnitude improvements over optimized standard search techniques and a Recurrent Neural Network-based approach to the problem.

In summary, we define and instantiate a framework for using deep learning for program synthesis problems like ones appearing on programming competition websites. Our concrete contributions are:

1. defining a programming language that is expressive enough to include real-world programming problems while being high-level enough to be predictable from input-output examples;

2. models for mapping sets of input-output examples to program properties; and

3. experiments that show an order of magnitude speedup over standard program synthesis techniques, which makes this approach feasible for solving problems of similar difficulty as the simplest problems that appear on programming competition websites.

## 2 Background on Inductive Program Synthesis

We begin by providing background on Inductive Program Synthesis, including a brief overview of how it is typically formulated and solved in the programming languages community.

The *Inductive Program Synthesis* (IPS) problem is the following: given input-output examples, produce a program that has behavior consistent with the examples.

Building an IPS system requires solving two problems. *First*, the search problem: to find consistent programs we need to search over a suitable set of possible programs. We need to define the set (i.e., the program space) and search procedure. *Second*, the ranking problem: if there are multiple programs consistent with the input-output examples, which one do we return? Both of these problems are dependent on the specifics of the problem formulation. Thus, the first important decision in formulating an approach to program synthesis is the choice of a *Domain Specific Language*.

**Domain Specific Languages (DSLs).** DSLs are programming languages that are suitable for a specialized domain but are more restrictive than full-featured programming languages. For example, one might disallow loops or other control flow, and only allow string data types and a small number of primitive operations like concatenation. Most of program synthesis research focuses on synthesizing programs in DSLs, because full-featured languages like C++ enlarge the search space and complicate synthesis. Restricted DSLs can also enable more efficient special-purpose search algorithms. For example, if a DSL only allows concatenations of substrings of an input string, a dynamic programming algorithm can efficiently search over all possible programs (Polozov & Gulwani, 2015). The choice of DSL also affects the difficulty of the ranking problem. For example, in a DSL without `if` statements, the same algorithm is applied to all inputs, reducing the number of programs consistent with any set of input-output examples, and thus the ranking problem becomes easier. Of course, the restrictiveness of the chosen DSL also determines which problems the system can solve at all.

**Search Techniques.** There are many techniques for searching for programs consistent with input-output examples. Perhaps the simplest approach is to define a grammar and then enumerate all derivations of the grammar, checking each one for consistency with the examples. This approach can be combined with pruning based on types and other logical reasoning (Feser et al., 2015). While simple, these approaches can be implemented efficiently, and they can be surprisingly effective.

In restricted domains such as the concatenation example discussed above, special-purpose algorithms can be used. FlashMeta (Polozov & Gulwani, 2015) describes a framework for DSLs which allow decomposition of the search problem, e.g., where the production of an output string from an input string can be reduced to finding a program for producing the first part of the output and concatenating it with a program for producing the latter part of the output string.

Another class of systems is based on Satisfiability Modulo Theories (SMT) solving. SMT combines SAT-style search with *theories* like arithmetic and inequalities, with the benefit that theory-dependent subproblems can be handled by special-purpose solvers. For example, a special-purpose solver can easily find integers $x, y$ such that $x < y$ and $y < -100$ hold, whereas an enumeration strategy may need to consider many values before satisfying the constraints. Many program synthesis engines based on SMT solvers exist, e.g., Sketch (Solar-Lezama, 2008) and Brahma (Gulwani et al., 2011). They convert the semantics of a DSL into a set of constraints between variables representing the

program and the input-output values, and then call an SMT solver to find a satisfying setting of the program variables. This approach shines when special-purpose reasoning can be leveraged, but complex DSLs can lead to very large constraint problems where constructing and manipulating the constraints can be a lot slower than an enumerative approach.

Finally, stochastic local search can be employed to search over program space, and there is a long history of applying genetic algorithms to this problem. One of the most successful recent examples is the STOKE super-optimization system (Schkufza et al., 2016), which uses stochastic local search to find assembly programs that have the same semantics as an input program but execute faster.

**Ranking.**     While we focus on the search problem in this work, we briefly mention the ranking problem here. A popular choice for ranking is to choose the shortest program consistent with input-output examples (Gulwani, 2016). A more sophisticated approach is employed by FlashFill (Singh & Gulwani, 2015). It works in a manner similar to max-margin structured prediction, where known ground truth programs are given, and the learning task is to assign scores to programs such that the ground truth programs score higher than other programs that satisfy the input-output specification.

## 3    LEARNING INDUCTIVE PROGRAM SYNTHESIS (LIPS)

In this section we outline the general approach that we follow in this work, which we call *Learning Inductive Program Synthesis* (LIPS). The details of our instantiation of LIPS appear in Sect. 4. The components of LIPS are (1) a DSL specification, (2) a data-generation procedure, (3) a machine learning model that maps from input-output examples to program attributes, and (4) a search procedure that searches program space in an order guided by the model from (3). The framework is related to the formulation of Menon et al. (2013); the relationship and key differences are discussed in Sect. 6.

**(1) DSL and Attributes.**     The choice of DSL is important in LIPS, just as it is in any program synthesis system. It should be expressive enough to capture the problems that we wish to solve, but restricted as much as possible to limit the difficulty of the search. In LIPS we additionally specify an *attribute function* $\mathcal{A}$ that maps programs $P$ of the DSL to finite *attribute vectors* $\boldsymbol{a} = \mathcal{A}(P)$. (Attribute vectors of different programs need not have equal length.) Attributes serve as the link between the machine learning and the search component of LIPS: the machine learning model predicts a distribution $q(\boldsymbol{a} \mid \mathcal{E})$, where $\mathcal{E}$ is the set of input-output examples, and the search procedure aims to search over programs $P$ as ordered by $q(\mathcal{A}(P) \mid \mathcal{E})$. Thus an attribute is useful if it is both predictable from input-output examples, and if conditioning on its value significantly reduces the effective size of the search space.

Possible attributes are the (perhaps position-dependent) presence or absence of high-level functions (e.g., does the program contain or end in a call to SORT). Other possible attributes include control flow templates (e.g., the number of loops and conditionals). In the extreme case, one may set $\mathcal{A}$ to the identity function, in which case the attribute is equivalent to the program; however, in our experiments we find that performance is improved by choosing a more abstract attribute function.

**(2) Data Generation.**     Step 2 is to generate a dataset $((P^{(n)}, \boldsymbol{a}^{(n)}, \mathcal{E}^{(n)}))_{n=1}^{N}$ of programs $P^{(n)}$ in the chosen DSL, their attributes $\boldsymbol{a}^{(n)}$, and accompanying input-output examples $\mathcal{E}^{(n)}$. Different approaches are possible, ranging from enumerating valid programs in the DSL and pruning, to training a more sophisticated generative model of programs in the DSL. The key in the LIPS formulation is to ensure that it is feasible to generate a large dataset (ideally millions of programs).

**(3) Machine Learning Model.**     The machine learning problem is to learn a distribution of attributes given input-output examples, $q(\boldsymbol{a} \mid \mathcal{E})$. There is freedom to explore a large space of models, so long as the input component can encode $\mathcal{E}$, and the output is a proper distribution over attributes (e.g., if attributes are a fixed-size binary vector, then a neural network with independent sigmoid outputs is appropriate; if attributes are variable size, then a recurrent neural network output could be used). Attributes are observed at training time, so training can use a maximum likelihood objective.

**(4) Search.**     The aim of the search component is to interface with an existing solver, using the predicted $q(\boldsymbol{a} \mid \mathcal{E})$ to guide the search. We describe specific approaches in the next section.

## 4 DEEPCODER

Here we describe DeepCoder, our instantiation of LIPS including a choice of DSL, a data generation strategy, models for encoding input-output sets, and algorithms for searching over program space.

### 4.1 DOMAIN SPECIFIC LANGUAGE AND ATTRIBUTES

We consider binary attributes indicating the presence or absence of high-level functions in the target program. To make this effective, the chosen DSL needs to contain constructs that are not so low-level that they all appear in the vast majority of programs, but at the same time should be common enough so that predicting their occurrence from input-output examples can be learned successfully.

Following this observation, our DSL is loosely inspired by query languages such as SQL or LINQ, where high-level functions are used in sequence to manipulate data. A program in our DSL is a sequence of function calls, where the result of each call initializes a fresh variable that is either a singleton integer or an integer array. Functions can be applied to any of the inputs or previously computed (intermediate) variables. The output of the program is the return value of the last function call, i.e., the last variable. See Fig. 1 for an example program of length $T = 4$ in our DSL.

```
a ← [int]          An input-output example:
b ← FILTER (<0) a   Input:
c ← MAP (*4) b      [-17, -3, 4, 11, 0, -5, -9, 13, 6, 6, -8, 11]
d ← SORT c          Output:
e ← REVERSE d       [-12, -20, -32, -36, -68]
```

Figure 1: An example program in our DSL that takes a single integer array as its input.

Overall, our DSL contains the first-order functions HEAD, LAST, TAKE, DROP, ACCESS, MINIMUM, MAXIMUM, REVERSE, SORT, SUM, and the higher-order functions MAP, FILTER, COUNT, ZIP-WITH, SCANL1. Higher-order functions require suitable lambda functions for their behavior to be fully specified: for MAP our DSL provides lambdas (+1), (-1), (*2), (/2), (*(-1)), (**2), (*3), (/3), (*4), (/4); for FILTER and COUNT there are predicates (>0), (<0), (%2==0), (%2==1) and for ZIPWITH and SCANL1 the DSL provides lambdas (+), (-), (*), MIN, MAX. A description of the semantics of all functions is provided in Appendix F.

Note that while the language only allows linear control flow, many of its functions do perform branching and looping internally (e.g., SORT, COUNT, ...). Examples of more sophisticated programs expressible in our DSL, which were inspired by the simplest problems appearing on programming competition websites, are shown in Appendix A.

### 4.2 DATA GENERATION

To generate a dataset, we enumerate programs in the DSL, heuristically pruning away those with easily detectable issues such as a redundant variable whose value does not affect the program output, or, more generally, existence of a shorter equivalent program (equivalence can be overapproximated by identical behavior on randomly or carefully chosen inputs). To generate valid inputs for a program, we enforce a constraint on the output value bounding integers to some predetermined range, and then propagate these constraints backward through the program to obtain a range of valid values for each input. If one of these ranges is empty, we discard the program. Otherwise, input-output pairs can be generated by picking inputs from the pre-computed valid ranges and executing the program to obtain the output values. The binary attribute vectors are easily computed from the program source codes.

### 4.3 MACHINE LEARNING MODEL

Observe how the input-output data in Fig. 1 is informative of the functions appearing in the program: the values in the output are all negative, divisible by 4, they are sorted in decreasing order, and they happen to be multiples of numbers appearing in the input. Our aim is to learn to recognize such patterns in the input-output examples, and to leverage them to predict the presence or absence of

individual functions. We employ neural networks to model and learn the mapping from input-output examples to attributes. We can think of these networks as consisting of two parts:

1. an *encoder*: a differentiable mapping from a set of $M$ input-output examples generated by a single program to a latent real-valued vector, and

2. a *decoder*: a differentiable mapping from the latent vector representing a set of $M$ input-output examples to predictions of the ground truth program's attributes.

For the encoder we use a simple feed-forward architecture. First, we represent the input and output types (singleton or array) by a one-hot-encoding, and we pad the inputs and outputs to a maximum length $L$ with a special NULL value. Second, each integer in the inputs and in the output is mapped to a learned embedding vector of size $E = 20$. (The range of integers is restricted to a finite range and each embedding is parametrized individually.) Third, for each input-output example separately, we concatenate the embeddings of the input types, the inputs, the output type, and the output into a single (fixed-length) vector, and pass this vector through $H = 3$ hidden layers containing $K = 256$ sigmoid units each. The third hidden layer thus provides an encoding of each individual input-output example. Finally, for input-output examples in a set generated from the same program, we pool these representations together by simple arithmetic averaging. See Appendix C for more details.

The advantage of this encoder lies in its simplicity, and we found it reasonably easy to train. A disadvantage is that it requires an upper bound $L$ on the length of arrays appearing in the input and output. We confirmed that the chosen encoder architecture is sensible in that it performs empirically at least as well as an RNN encoder, a natural baseline, which may however be more difficult to train.

DeepCoder learns to predict presence or absence of individual functions of the DSL. We shall see this can already be exploited by various search techniques to large computational gains. We use a decoder that pre-multiplies the encoding of input-output examples by a learned $C \times K$ matrix, where $C = 34$ is the number of functions in our DSL (higher-order functions and lambdas are predicted independently), and treats the resulting $C$ numbers as log-unnormalized probabilities (logits) of each function appearing in the source code. Fig. 2 shows the predictions a trained neural network made from 5 input-output examples for the program shown in Fig. 1.

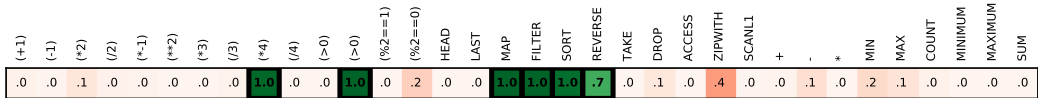

Figure 2: Neural network predicts the probability of each function appearing in the source code.

## 4.4 SEARCH

One of the central ideas of this work is to use a neural network to guide the search for a program consistent with a set of input-output examples instead of directly predicting the entire source code. This section briefly describes the search techniques and how they integrate the predicted attributes.

**Depth-first search (DFS).**    We use an optimized version of DFS to search over programs with a given maximum length $T$ (see Appendix D for details). When the search procedure extends a partial program by a new function, it has to try the functions in the DSL in some order. At this point DFS can opt to consider the functions as ordered by their predicted probabilities from the neural network.

**"Sort and add" enumeration.**    A stronger way of utilizing the predicted probabilities of functions in an enumerative search procedure is to use a *Sort and add* scheme, which maintains a set of *active* functions and performs DFS with the active function set only. Whenever the search fails, the next most probable function (or several) are added to the active set and the search restarts with this larger active set. Note that this scheme has the deficiency of potentially re-exploring some parts of the search space several times, which could be avoided by a more sophisticated search procedure.

**Sketch.**    Sketch (Solar-Lezama, 2008) is a successful SMT-based program synthesis tool from the programming languages research community. While its main use case is to synthesize programs

by filling in "holes" in incomplete source code so as to match specified requirements, it is flexible enough for our use case as well. The function in each step and its arguments can be treated as the "holes", and the requirement to be satisfied is consistency with the provided set of input-output examples. Sketch can utilize the neural network predictions in a *Sort and add* scheme as described above, as the possibilities for each function hole can be restricted to the current active set.

$\lambda^2$. $\lambda^2$ (Feser et al., 2015) is a program synthesis tool from the programming languages community that combines enumerative search with deduction to prune the search space. It is designed to infer small functional programs for data structure manipulation from input-output examples, by combining functions from a provided library. $\lambda^2$ can be used in our framework using a *Sort and add* scheme as described above by choosing the library of functions according to the neural network predictions.

### 4.5 TRAINING LOSS FUNCTION

We use the negative cross entropy loss to train the neural network described in Sect. 4.3, so that its predictions about each function can be interpreted as marginal probabilities. The LIPS framework dictates learning $q(\boldsymbol{a} \mid \mathcal{E})$, the joint distribution of all attributes $\boldsymbol{a}$ given the input-output examples, and it is not clear a priori how much DeepCoder loses by ignoring correlations between functions. However, under the simplifying assumption that the runtime of searching for a program of length $T$ with $C$ functions made available to a search routine is proportional to $C^T$, the following result for *Sort and add* procedures shows that their runtime can be optimized using marginal probabilities.

**Lemma 1.** *For any fixed program length $T$, the expected total runtime of a* Sort and add *search scheme can be upper bounded by a quantity that is minimized by adding the functions in the order of decreasing true marginal probabilities.*

*Proof.* Predicting source code functions from input-output examples can be seen as a multi-label classification problem, where each set of input-output examples is associated with a set of relevant labels (functions appearing in the ground truth source code). Dembczynski et al. (2010) showed that in multi-label classification under a so-called *Rank loss*, it is Bayes optimal to rank the labels according to their marginal probabilities. If the runtime of search with $C$ functions is proportional to $C^T$, the total runtime of a *Sort and add* procedure can be monotonically transformed so that it is upper bounded by this Rank loss. See Appendix E for more details. □

## 5 EXPERIMENTS

In this section we report results from two categories of experiments. Our main experiments (Sect. 5.1) show that the LIPS framework can lead to significant performance gains in solving IPS by demonstrating such gains with DeepCoder. In Sect. 5.2 we illustrate the robustness of the method by demonstrating a strong kind of generalization ability across programs of different lengths.

### 5.1 DEEPCODER COMPARED TO BASELINES

We trained a neural network as described in Sect. 4.3 to predict used functions from input-output examples and constructed a test set of $P = 500$ programs, guaranteed to be semantically disjoint from all programs on which the neural network was trained (similarly to the equivalence check described in Sect. 4.2, we have ensured that all test programs behave differently from all programs used during training on at least one input). For each test program we generated $M = 5$ input-output examples involving integers of magnitudes up to 256, passed the examples to the trained neural network, and fed the obtained predictions to the search procedures from Sect. 4.4. We also considered a RNN-based decoder generating programs using beam search (see Sect. 5.3 for details). To evaluate DeepCoder, we then recorded the time the search procedures needed to find a program consistent with the $M$ input-output examples. As a baseline, we also ran all search procedures using a simple prior as function probabilities, computed from their global incidence in the program corpus.

In the first, smaller-scale experiment (program search space size $\sim 2 \times 10^6$) we trained the neural network on programs of length $T = 3$, and the test programs were of the same length. Table 1 shows the per-task timeout required such that a solution could be found for given proportions of the test tasks (in time less than or equal to the timeout). For example, in a hypothetical test set with 4 tasks

Table 1: Search speedups on programs of length $T = 3$ due to using neural network predictions.

| Timeout needed | DFS | | | Enumeration | | | $\lambda^2$ | | | Sketch | | Beam |
|---|---|---|---|---|---|---|---|---|---|---|---|---|
| to solve | 20% | 40% | 60% | 20% | 40% | 60% | 20% | 40% | 60% | 20% | 40% | 20% |
| Baseline | $41ms$ | $126ms$ | $314ms$ | $80ms$ | $335ms$ | $861ms$ | $18.9s$ | $49.6s$ | $84.2s$ | $>10^3s$ | $>10^3s$ | $>10^3s$ |
| DeepCoder | $2.7ms$ | $33ms$ | $110ms$ | $1.3ms$ | $6.1ms$ | $27ms$ | $0.23s$ | $0.52s$ | $13.5s$ | $2.13s$ | $455s$ | $292s$ |
| Speedup | $15.2\times$ | $3.9\times$ | $2.9\times$ | $62.2\times$ | $54.6\times$ | $31.5\times$ | $80.4\times$ | $94.6\times$ | $6.2\times$ | $>467\times$ | $>2.2\times$ | $>3.4\times$ |

and runtimes of $3s$, $2s$, $1s$, $4s$, the timeout required to solve 50% of tasks would be $2s$. More detailed experimental results are discussed in Appendix B.

In the main experiment, we tackled a large-scale problem of searching for programs consistent with input-output examples generated from programs of length $T = 5$ (search space size on the order of $10^{10}$), supported by a neural network trained with programs of shorter length $T = 4$. Here, we only consider $P = 100$ programs for reasons of computational efficiency, after having verified that this does not significantly affect the results in Table 1. The table in Fig. 3a shows significant speedups for DFS, *Sort and add* enumeration, and $\lambda^2$ with *Sort and add* enumeration, the search techniques capable of solving the search problem in reasonable time frames. Note that *Sort and add* enumeration without the neural network (using prior probabilities of functions) exceeded the $10^4$ second timeout in two cases, so the relative speedups shown are crude lower bounds.

| Timeout needed | DFS | | | Enumeration | | | $\lambda^2$ |
|---|---|---|---|---|---|---|---|
| to solve | 20% | 40% | 60% | 20% | 40% | 60% | 20% |
| Baseline | $163s$ | $2887s$ | $6832s$ | $8181s$ | $>10^4s$ | $>10^4s$ | $463s$ |
| DeepCoder | $24s$ | $514s$ | $2654s$ | $9s$ | $264s$ | $4640s$ | $48s$ |
| Speedup | $6.8\times$ | $5.6\times$ | $2.6\times$ | $907\times$ | $>37\times$ | $>2\times$ | $9.6\times$ |

(a)

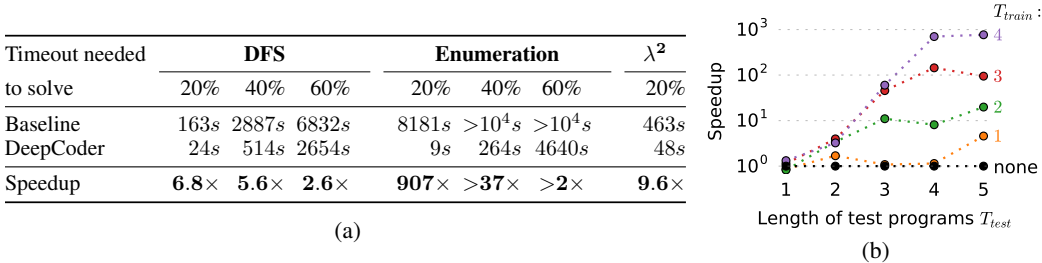

(b)

Figure 3: Search speedups on programs of length $T = 5$ and influence of length of training programs.

We hypothesize that the substantially larger performance gains on *Sort and add* schemes as compared to gains on DFS can be explained by the fact that the choice of attribute function (predicting presence of functions anywhere in the program) and learning objective of the neural network are better matched to the *Sort and add* schemes. Indeed, a more appropriate attribute function for DFS would be one that is more informative of the functions appearing early in the program, since exploring an incorrect first function is costly with DFS. On the other hand, the discussion in Sect. 4.5 provides theoretical indication that ignoring the correlations between functions is not cataclysmic for *Sort and add* enumeration, since a Rank loss that upper bounds the *Sort and add* runtime can still be minimized.

In Appendix G we analyse the performance of the neural networks used in these experiments, by investigating which attributes (program instructions) tend to be difficult to distinguish from each other.

## 5.2 GENERALIZATION ACROSS PROGRAM LENGTHS

To investigate the encoder's generalization ability across programs of different lengths, we trained a network to predict used functions from input-output examples that were generated from programs of length $T_{\text{train}} \in \{1, \ldots, 4\}$. We then used each of these networks to predict functions on 5 test sets containing input-output examples generated from programs of lengths $T_{\text{test}} \in \{1, \ldots, 5\}$, respectively. The test programs of a given length $T$ were semantically disjoint from all training programs of the same length $T$ and also from all training and test programs of shorter lengths $T' < T$.

For each of the combinations of $T_{\text{train}}$ and $T_{\text{test}}$, *Sort and add* enumerative search was run both with and without using the neural network's predictions (in the latter case using prior probabilities) until it solved 20% of the test set tasks. Fig. 3b shows the relative speedup of the solver having access to predictions from the trained neural networks. These results indicate that the neural networks are able to generalize beyond programs of the same length that they were trained on. This is partly due to the

search procedure on top of their predictions, which has the opportunity to correct for the presence of functions that the neural network failed to predict. Note that a sequence-to-sequence model trained on programs of a fixed length could not be expected to exhibit this kind of generalization ability.

## 5.3 ALTERNATIVE MODELS

**Encoder**  We evaluated replacing the feed-forward architecture encoder (Sect. 4.3) with an RNN, a natural baseline. Using a GRU-based RNN we were able to achieve results almost as good as using the feed-forward architecture, but found the RNN encoder more difficult to train.

**Decoder**  We also considered a purely neural network-based approach, where an RNN decoder is trained to predict the entire program token-by-token. We combined this with our feed-forward encoder by initializing the RNN using the pooled final layer of the encoder. We found it substantially more difficult to train an RNN decoder as compared to the independent binary classifiers employed above. Beam search was used to explore likely programs predicted by the RNN, but it only lead to a solution comparable with the other techniques when searching for programs of lengths $T \leq 2$, where the search space size is very small (on the order of $10^3$). Note that using an RNN for both the encoder and decoder corresponds to a standard sequence-to-sequence model. However, we do do not rule out that a more sophisticated RNN decoder or training procedure could be possibly more successful.

## 6 RELATED WORK

**Machine Learning for Inductive Program Synthesis.**  There is relatively little work on using machine learning for programming by example. The most closely related work is that of Menon et al. (2013), in which a hand-coded set of features of input-output examples are used as "clues." When a clue appears in the input-output examples (e.g., the output is a permutation of the input), it reweights the probabilities of productions in a probabilistic context free grammar by a learned amount. This work shares the idea of learning to guide the search over program space conditional on input-output examples. One difference is in the domains. Menon et al. (2013) operate on short string manipulation programs, where it is arguably easier to hand-code features to recognize patterns in the input-output examples (e.g., if the outputs are always permutations or substrings of the input). Our work shows that there are strong cues in patterns in input-output examples in the domain of numbers and lists. However, the main difference is the scale. Menon et al. (2013) learns from a small (280 examples), manually-constructed dataset, which limits the capacity of the machine learning model that can be trained. Thus, it forces the machine learning component to be relatively simple. Indeed, Menon et al. (2013) use a log-linear model and rely on hand-constructed features. LIPS automatically generates training data, which yields datasets with millions of programs and enables high-capacity deep learning models to be brought to bear on the problem.

**Learning Representations of Program State.**  Piech et al. (2015) propose to learn joint embeddings of program states and programs to automatically extend teacher feedback to many similar programs in the MOOC setting. This work is similar in that it considers embedding program states, but the domain is different, and it otherwise specifically focuses on syntactic differences between semantically equivalent programs to provide stylistic feedback. Li et al. (2016) use graph neural networks (GNNs) to predict logical descriptions from program states, focusing on data structure shapes instead of numerical and list data. Such GNNs may be a suitable architecture to encode states appearing when extending our DSL to handle more complex data structures.

**Learning to Infer.**  Very recently, Alemi et al. (2016) used neural sequence models in tandem with an automated theorem prover. Similar to our *Sort and Add* strategy, a neural network component is trained to select premises that the theorem prover can use to prove a theorem. A recent extension (Loos et al., 2017) is similar to our DFS enumeration strategy and uses a neural network to guide the proof search at intermediate steps. The main differences are in the domains, and that they train on an existing corpus of theorems. More broadly, if we view a DSL as defining a model and search as a form of inference algorithm, then there is a large body of work on using discriminatively-trained models to aid inference in generative models. Examples include Dayan et al. (1995); Kingma & Welling (2014); Shotton et al. (2013); Stuhlmüller et al. (2013); Heess et al. (2013); Jampani et al. (2015).

## 7   DISCUSSION AND FUTURE WORK

We have presented a framework for improving IPS systems by using neural networks to translate cues in input-output examples to guidance over where to search in program space. Our empirical results show that for many programs, this technique improves the runtime of a wide range of IPS baselines by 1-3 orders. We have found several problems in real online programming challenges that can be solved with a program in our language, which validates the relevance of the class of problems that we have studied in this work. In sum, this suggests that we have made significant progress towards being able to solve programming competition problems, and the machine learning component plays an important role in making it tractable.

There remain some limitations, however. First, the programs we can synthesize are only the simplest problems on programming competition websites and are simpler than most competition problems. Many problems require more complex algorithmic solutions like dynamic programming and search, which are currently beyond our reach. Our chosen DSL currently cannot express solutions to many problems. To do so, it would need to be extended by adding more primitives and allow for more flexibility in program constructs (such as allowing loops). Second, we currently use five input-output examples with relatively large integer values (up to 256 in magnitude), which are probably more informative than typical (smaller) examples. While we remain optimistic about LIPS's applicability as the DSL becomes more complex and the input-output examples become less informative, it remains to be seen what the magnitude of these effects are as we move towards solving large subsets of programming competition problems.

We foresee many extensions of DeepCoder. We are most interested in better data generation procedures by using generative models of source code, and to incorporate natural language problem descriptions to lessen the information burden required from input-output examples. In sum, Deep-Coder represents a promising direction forward, and we are optimistic about the future prospects of using machine learning to synthesize programs.

### ACKNOWLEDGMENTS

The authors would like to express their gratitude to Rishabh Singh and Jack Feser for their valuable guidance and help on using the Sketch and $\lambda^2$ program synthesis systems.

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

## A  EXAMPLE PROGRAMS

This section shows example programs in our Domain Specific Language (DSL), together with input-output examples and short descriptions. These programs have been inspired by simple tasks appearing on real programming competition websites, and are meant to illustrate the expressive power of our DSL.

| **Program 0:** | **Input-output example:** | *Description*: |
|---|---|---|
| k ← int | *Input*: | A new shop near you is selling $n$ paintings. You have $k < n$ friends and you would like to buy each of your friends a painting from the shop. Return the minimal amount of money you will need to spend. |
| b ← [int] | 2, [3 5 4 7 5] | |
| c ← SORT b | *Output*: | |
| d ← TAKE k c | [7] | |
| e ← SUM d | | |

| **Program 1:** | **Input-output example:** | *Description*: |
|---|---|---|
| w ← [int] | *Input*: | In soccer leagues, match winners are awarded 3 points, losers 0 points, and both teams get 1 point in the case of a tie. Compute the number of points awarded to the winner of a league given two arrays $w, t$ of the same length, where $w[i]$ (resp. $t[i]$) is the number of times team $i$ won (resp. tied). |
| t ← [int] | [6 2 4 7 9], | |
| c ← MAP (*3) w | [5 3 6 1 0] | |
| d ← ZIPWITH (+) c t | *Output*: | |
| e ← MAXIMUM d | 27 | |

| **Program 2:** | **Input-output example:** | *Description*: |
|---|---|---|
| a ← [int] | *Input*: | Alice and Bob are comparing their results in a recent exam. Given their marks per question as two arrays $a$ and $b$, count on how many questions Alice got more points than Bob. |
| b ← [int] | [6 2 4 7 9], | |
| c ← ZIPWITH (−) b a | [5 3 2 1 0] | |
| d ← COUNT (>0) c | *Output*: | |
| | 4 | |

| **Program 3:** | **Input-output example:** | *Description*: |
|---|---|---|
| h ← [int] | *Input*: | Perditia is very peculiar about her garden and wants that the trees standing in a row are all of non-increasing heights. Given the tree heights in centimeters in order of the row as an array h, compute how many centimeters she needs to trim the trees in total. |
| b ← SCANL1 MIN h | [8 5 7 2 5] | |
| c ← ZIPWITH (−) h b | *Output*: | |
| d ← FILTER (>0) c | 5 | |
| e ← SUM d | | |

| **Program 4:** | **Input-output example:** | *Description*: |
|---|---|---|
| x ← [int] | *Input*: | Xavier and Yasmine are laying sticks to form non-overlapping rectangles on the ground. They both have fixed sets of pairs of sticks of certain lengths (represented as arrays x and y of numbers). Xavier only lays sticks parallel to the x axis, and Yasmine lays sticks only parallel to y axis. Compute the area their rectangles will cover at least. |
| y ← [int] | [7 3 8 2 5], | |
| c ← SORT x | [2 8 9 1 3] | |
| d ← SORT y | *Output*: | |
| e ← REVERSE d | 79 | |
| f ← ZIPWITH (*) d e | | |
| g ← SUM f | | |

| **Program 5:** | **Input-output example:** | *Description*: |
|---|---|---|
| a ← [int] | *Input*: | A sequence called Billy is looking into the mirror, wondering how much weight it could lose by replacing any of its elements by their mirror images. Given a description of Billy as an array $b$ of length $n$, return an array $c$ of minimal sum where each element $c[i]$ is either $b[i]$ or its mirror image $b[n - i - 1]$. |
| b ← REVERSE a | [3 7 5 2 8] | |
| c ← ZIPWITH MIN a b | *Output*: | |
| | [3 2 5 2 3] | |

| **Program 6:** | **IO example:** | *Description*: |
|---|---|---|
| t ← [int] | *Input*: | Umberto has a large collection of ties and match- |
| p ← [int] | [4 8 11 2], | ing pocket squares—too large, his wife says—and he |
| c ← MAP (-1) t | [2 3 4 1] | needs to sell one pair. Given their values as arrays t |
| d ← MAP (-1) p | *Output*: | and p, assuming that he sells the cheapest pair, and |
| e ← ZIPWITH (+) c d | 1 | selling costs 2, how much will he lose from the sale? |
| f ← MINIMUM e | | |

| | | *Description*: |
|---|---|---|
| | | Zack always promised his $n$ friends to buy them |
| | | candy, but never did. Now he won the lottery |
| | | and counts how often and how much candy he |
| **Program 7:** | **IO example:** | promised to his friends, obtaining arrays p (num- |
| s ← [int] | *Input*: | ber of promises) and s (number of promised sweets). |
| p ← [int] | [4 7 2 3], | He announces that to repay them, he will buy |
| c ← SCANL1 (+) p | [2 1 3 1] | s[1]+s[2]+...+s[n] pieces of candy for the |
| d ← ZIPWITH (*) s c | *Output*: | first p[1] days, then s[2]+s[3]+...+s[n] for |
| e ← SUM d | 62 | p[2] days, and so on, until he has fulfilled all |
| | | promises. How much candy will he buy in total? |

| | | *Description*: |
|---|---|---|
| | | Vivian loves rearranging things. Most of all, when |
| | | she sees a row of heaps, she wants to make sure that |
| | | each heap has more items than the one to its left. She |
| | | is also obsessed with efficiency, so always moves the |
| **Program 8:** | **IO example:** | least possible number of items. Her dad really dislikes |
| s ← [int] | *Input*: | if she changes the size of heaps, so she only moves |
| b ← REVERSE s | [1 2 4 5 7] | single items between them, making sure that the set of |
| c ← ZIPWITH (-) b s | *Output*: | sizes of the heaps is the same as at the start; they are |
| d ← FILTER (>0) c | 9 | only in a different order. When you come in, you see |
| e ← SUM d | | heaps of sizes (of course, sizes strictly monotonically |
| | | increasing) s[0], s[1], ... s[n]. What is |
| | | the maximal number of items that Vivian could have |
| | | moved? |

Fig. 4 shows the predictions made by a neural network trained on programs of length $T = 4$ that were ensured to be semantically disjoint from all 9 example programs shown in this section. For each task, the neural network was provided with 5 input-output examples.

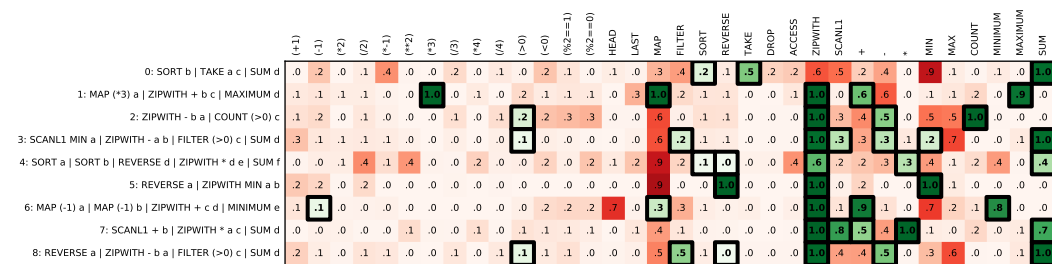

Figure 4: Predictions of a neural network on the 9 example programs described in this section. Numbers in squares would ideally be close to 1 (function is present in the ground truth source code), whereas all other numbers should ideally be close to 0 (function is not needed).

# B EXPERIMENTAL RESULTS

Results presented in Sect. 5.1 showcased the computational speedups obtained from the LIPS frame-work (using DeepCoder), as opposed to solving each program synthesis problem with only the

information about global incidence of functions in source code available. For completeness, here we show plots of raw computation times of each search procedure to solve a given number of problems.

Fig. 5 shows the computation times of DFS, of Enumerative search with a *Sort and add* scheme, of the $\lambda^2$ and Sketch solvers with a *Sort and add* scheme, and of Beam search, when searching for a program consistent with input-output examples generated from $P = 500$ different test programs of length $T = 3$. As discussed in Sect. 5.1, these test programs were ensured to be semantically disjoint from all programs used to train the neural networks, as well as from all programs of shorter length (as discussed in Sect. 4.2).

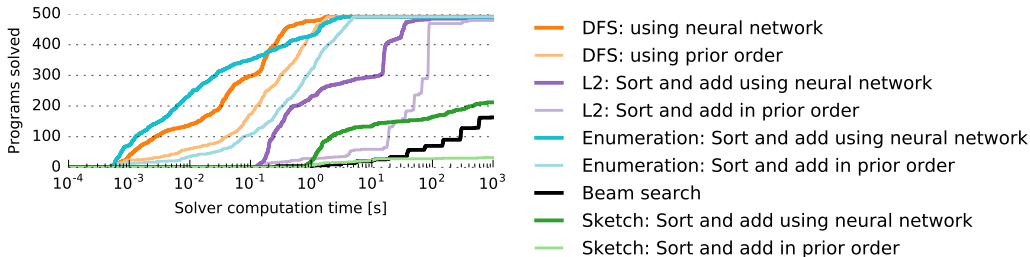

Figure 5: Number of test problems solved versus computation time.

The "steps" in the results for Beam search are due to our search strategy, which doubles the size of the considered beam until reaching the timeout (of 1000 seconds) and thus steps occur whenever the search for a beam of size $2^k$ is finished. For $\lambda^2$, we observed that no solution for a given set of allowed functions was ever found after about 5 seconds (on the benchmark machines), but that $\lambda^2$ continued to search. Hence, we introduced a hard timeout after 6 seconds for all but the last iterations of our *Sort and add* scheme.

Fig. 6 shows the computation times of DFS, Enumerative search with a *Sort and add* scheme, and $\lambda^2$ with a *Sort and add* scheme when searching for programs consistent with input-output examples generated from $P = 100$ different test programs of length $T = 5$. The neural network was trained on programs of length $T = 4$.

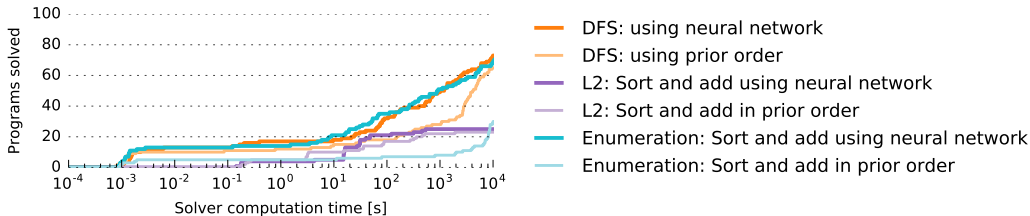

Figure 6: Number of test problems solved versus computation time.

## C  THE NEURAL NETWORK

As briefly described in Sect. 4.3, we used the following simple feed-forward architecture encoder:

- For each input-output example in the set generated from a single ground truth program:
  - Pad arrays appearing in the inputs and in the output to a maximum length $L = 20$ with a special NULL value.
  - Represent the type (singleton integer or integer array) of each input and of the output using a one-hot-encoding vector. Embed each integer in the valid integer range ($-256$ to $255$) using a learned embedding into $E = 20$ dimensional space. Also learn an embedding for the padding NULL value.

- – Concatenate the representations of the input types, the embeddings of integers in the inputs, the representation of the output type, and the embeddings of integers in the output into a single (fixed-length) vector.
- – Pass this vector through $H = 3$ hidden layers containing $K = 256$ sigmoid units each.
- Pool the last hidden layer encodings of each input-output example together by simple arithmetic averaging.

Fig. 7 shows a schematic drawing of this encoder architecture, together with the decoder that performs independent binary classification for each function in the DSL, indicating whether or not it appears in the ground truth source code.

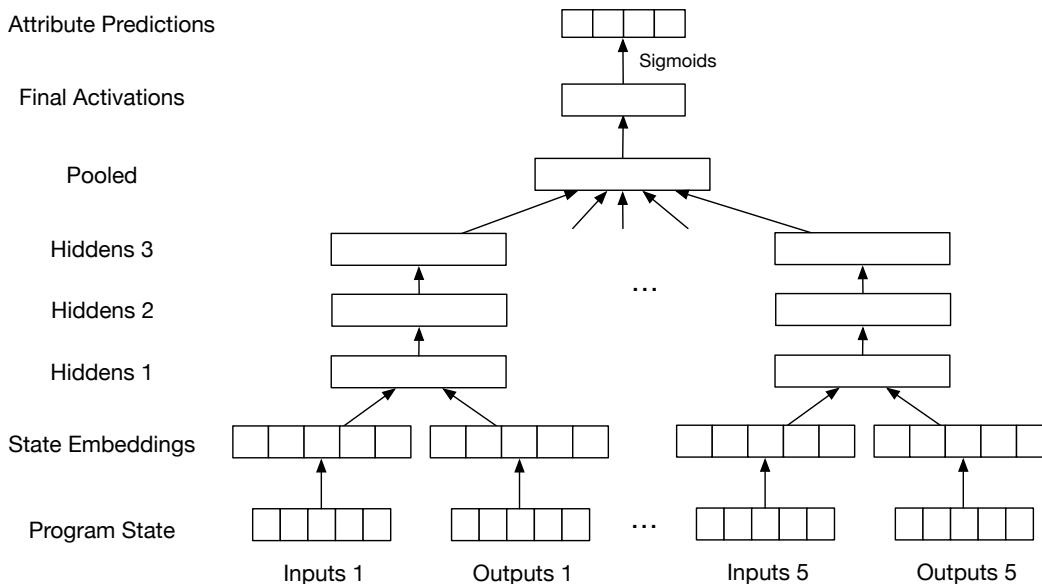

Figure 7: Schematic representation of our feed-forward encoder, and the decoder.

While DeepCoder learns to embed integers into a $E = 20$ dimensional space, we built the system up gradually, starting with a $E = 2$ dimensional space and only training on programs of length $T = 1$. Such a small scale setting allowed easier investigation of the workings of the neural network, and indeed Fig. 8 below shows a learned embedding of integers in $\mathbb{R}^2$. The figure demonstrates that the network has learnt the concepts of number magnitude, sign (positive or negative) and evenness, presumably due to FILTER (>0), FILTER (<0), FILTER (%2==0) and FILTER (%2==1) all being among the programs on which the network was trained.

# D  DEPTH-FIRST SEARCH

We use an optimized C++ implementation of depth-first search (DFS) to search over programs with a given maximum length $T$. In depth-first search, we start by choosing the first function (and its arguments) of a potential solution program, and then recursively consider all ways of filling in the rest of the program (up to length $T$), before moving on to a next choice of first instruction (if a solution has not yet been found).

A program is considered a solution if it is consistent with all $M = 5$ provided input-output examples. Note that this requires evaluating all candidate programs on the $M$ inputs and checking the results for equality with the provided $M$ respective outputs. Our implementation of DFS exploits the sequential structure of programs in our DSL by caching the results of evaluating all prefixes of the currently considered program on the example inputs, thus allowing efficient reuse of computation between candidate programs with common prefixes.

This allows us to explore the search space at roughly the speed of $\sim 3 \times 10^6$ programs per second.

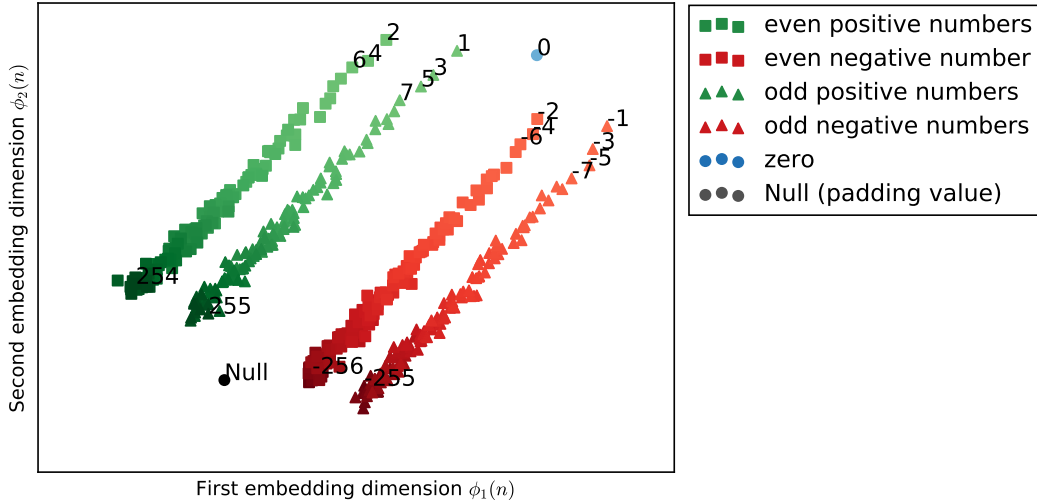

Figure 8: A learned embedding of integers $\{-256, -255, \ldots, -1, 0, 1, \ldots, 255\}$ in $\mathbb{R}^2$. The color intensity corresponds to the magnitude of the embedded integer.

When the search procedure extends a partial program by a new function, it has to try the functions in the DSL in some order. At this point DFS can opt to consider the functions as ordered by their predicted probabilities from the neural network. The probability of a function consisting of a higher-order function and a lambda is taken to be the minimum of the probabilities of the two constituent functions.

## E  TRAINING LOSS FUNCTION

In Sect. 4.5 we outlined a justification for using marginal probabilities of individual functions as a sensible intermediate representation to provide a solver employing a *Sort and add* scheme (we considered Enumerative search and the Sketch solver with this scheme). Here we provide a more detailed discussion.

Predicting program components from input-output examples can be cast as a multilabel classification problem, where each instance (set of input-output examples) is associated with a set of relevant labels (functions appearing in the code that generated the examples). We denote the number of labels (functions) by $C$, and note that throughout this work $C = 34$.

When the task is to predict a subset of labels $\mathbf{y} \in \{0, 1\}^C$, different loss functions can be employed to measure the prediction error of a classifier $\mathbf{h}(\mathbf{x})$ or ranking function $\mathbf{f}(\mathbf{x})$. Dembczynski et al. (2010) discuss the following three loss functions:

- *Hamming loss* counts the number of labels that are predicted incorrectly by a classifier $\mathbf{h}$:

$$L_H(\mathbf{y}, \mathbf{h}(\mathbf{x})) = \sum_{c=1}^{C} \mathbb{1}_{\{y_c \neq h_c(\mathbf{x})\}}$$

- *Rank loss* counts the number of label pairs violating the condition that relevant labels are ranked higher than irrelevant ones by a scoring function $\mathbf{f}$:

$$L_r(\mathbf{y}, \mathbf{f}(\mathbf{x})) = \sum_{(i,j):y_i=1,y_j=0} \mathbb{1}_{\{f_i < f_j\}}$$

- *Subset Zero-One loss* indicates whether all labels have been correctly predicted by $\mathbf{h}$:

$$L_s(\mathbf{y}, \mathbf{h}(\mathbf{x})) = \mathbb{1}_{\{\mathbf{y} \neq \mathbf{h}(\mathbf{x})\}}$$

Dembczynski et al. (2010) proved that Bayes optimal decisions under the Hamming and Rank loss functions, i.e., decisions minimizing the expected loss under these loss functions, can be computed from marginal probabilities $p_c(y_c|\mathbf{x})$. This *suggests* that:

- Multilabel classification under these two loss functions may not benefit from considering dependencies between the labels.

- "Instead of minimizing the Rank loss directly, one can simply use any approach for single label prediction that properly estimates the marginal probabilities." (Dembczyński et al., 2012)

Training the neural network with the negative cross entropy loss function as the training objective is precisely a method for properly estimating the marginal probabilities of labels (functions appearing in source code). It is thus a sensible step in preparation for making predictions under a Rank loss.

It remains to discuss the relationship between the Rank loss and the actual quantity we care about, which is the total runtime of a *Sort and add* search procedure. Recall the simplifying assumption that the runtime of searching for a program of length $T$ with $C$ functions made available to the search is proportional to $C^T$, and consider a *Sort and add* search for a program of length $T$, where the size of the active set is increased by 1 whenever the search fails. Starting with an active set of size 1, the total time until a solution is found can be upper bounded by

$$1^T + 2^T + \cdots + C_A^T \leq C_A^{T+1} \leq CC_A^T$$

where $C_A$ is the size of the active set when the search finally succeeds (i.e., when the active set finally contains all necessary functions for a solution to exist). Hence the total runtime of a *Sort and add* search can be upper bounded by a quantity that is proportional to $C_A^T$.

Now fix a valid program solution $P$ that requires $C_P$ functions, and let $\mathbf{y}_P \in \{0,1\}^C$ be the indicator vector of functions used by $P$. Let $D := C_A - C_P$ be the number of redundant operations added into the active set until all operations from $P$ have been added.

**Example 1.** Suppose the labels, as sorted by decreasing predicted marginal probabilities $\mathbf{f}(\mathbf{x})$, are as follows:

$$1\ 1\ 1\ 1\ {\color{red}0\ 0}\ 1\ {\color{red}0\ 0\ 0}\ {\color{green}1}\ 0\ 0\ 0\ 0\ 0\ 0\ 0\ 0\ 0\ 0\ 0\ 0\ 0\ 0\ 0\ 0\ 0\ 0\ 0\ 0\ 0\ 0$$

Then the solution $P$ contains $C_P = 6$ functions, but the active set needs to grow to size $C_A = 11$ to include all of them, adding $D = 5$ redundant functions along the way. Note that the rank loss of the predictions $\mathbf{f}(\mathbf{x})$ is $L_r(\mathbf{y}_P, \mathbf{f}(\mathbf{x})) = 2 + 5 = 7$, as it double counts the two redundant functions which are scored higher than two relevant labels.

Noting that in general $L_r(\mathbf{y}_P, \mathbf{f}(\mathbf{x})) \geq D$, the previous upper bound on the runtime of *Sort and add* can be further upper bounded as follows:

$$C_A^T = (C_P + D)^T \leq \text{const} + \text{const} \times D^T \leq \text{const} + \text{const} \times L_r(\mathbf{y}_P, \mathbf{f}(\mathbf{x}))^T$$

Hence we see that for a constant value of $T$, this upper bound can be minimized by optimizing the Rank loss of the predictions $\mathbf{f}(\mathbf{x})$. Note also that $L_r(\mathbf{y}_P, \mathbf{f}(\mathbf{x})) = 0$ would imply $D = 0$, in which case $C_A = C_P$.

# F   DOMAIN SPECIFIC LANGUAGE OF DEEPCODER

Here we provide a description of the semantics of our DSL from Sect. 4.1, both in English and as a Python implementation. Throughout, NULL is a special value that can be set e.g. to an integer outside the working integer range.

First-order functions:

- HEAD :: [int] -> int
  ```
  lambda xs: xs[0] if len(xs)>0 else Null
  ```
  Given an array, returns its first element (or NULL if the array is empty).

- LAST :: [int] -> int
  ```
  lambda xs: xs[-1] if len(xs)>0 else Null
  ```
  Given an array, returns its last element (or NULL if the array is empty).

- TAKE :: `int -> [int] -> int`
  `lambda n, xs: xs[:n]`
  Given an integer `n` and array `xs`, returns the array truncated after the `n`-th element. (If the length of `xs` was no larger than `n` in the first place, it is returned without modification.)

- DROP :: `int -> [int] -> int`
  `lambda n, xs: xs[n:]`
  Given an integer `n` and array `xs`, returns the array with the first `n` elements dropped. (If the length of `xs` was no larger than `n` in the first place, an empty array is returned.)

- ACCESS :: `int -> [int] -> int`
  `lambda n, xs: xs[n] if n>=0 and len(xs)>n else Null`
  Given an integer `n` and array `xs`, returns the (n+1)-st element of `xs`. (If the length of `xs` was less than or equal to `n`, the value NULL is returned instead.)

- MINIMUM :: `[int] -> int`
  `lambda xs: min(xs) if len(xs)>0 else Null`
  Given an array, returns its minimum (or NULL if the array is empty).

- MAXIMUM :: `[int] -> int`
  `lambda xs: max(xs) if len(xs)>0 else Null`
  Given an array, returns its maximum (or NULL if the array is empty).

- REVERSE :: `[int] -> [int]`
  `lambda xs: list(reversed(xs))`
  Given an array, returns its elements in reversed order.

- SORT :: `[int] -> [int]`
  `lambda xs: sorted(xs)`
  Given an array, return its elements in non-decreasing order.

- SUM :: `[int] -> int`
  `lambda xs: sum(xs)`
  Given an array, returns the sum of its elements. (The sum of an empty array is 0.)

Higher-order functions:

- MAP :: `(int -> int) -> [int] -> [int]`
  `lambda f, xs: [f(x) for x in xs]`
  Given a lambda function `f` mapping from integers to integers, and an array `xs`, returns the array resulting from applying `f` to each element of `xs`.

- FILTER :: `(int -> bool) -> [int] -> [int]`
  `lambda f, xs: [x for x in xs if f(x)]`
  Given a predicate `f` mapping from integers to truth values, and an array `xs`, returns the elements of `xs` satisfying the predicate in their original order.

- COUNT :: `(int -> bool) -> [int] -> int`
  `lambda f, xs: len([x for x in xs if f(x)])`
  Given a predicate `f` mapping from integers to truth values, and an array `xs`, returns the number of elements in `xs` satisfying the predicate.

- ZIPWITH :: `(int -> int -> int) -> [int] -> [int] -> [int]`
  `lambda f, xs, ys: [f(x, y) for (x, y) in zip(xs, ys)]`
  Given a lambda function `f` mapping integer pairs to integers, and two arrays `xs` and `ys`, returns the array resulting from applying `f` to corresponding elements of `xs` and `ys`. The length of the returned array is the minimum of the lengths of `xs` and `ys`.

- SCANL1 :: `(int -> int -> int) -> [int] -> [int]`
  Given a lambda function `f` mapping integer pairs to integers, and an array `xs`, returns an array `ys` of the same length as `xs` and with its content defined by the recurrence `ys[0] = xs[0]`, `ys[n] = f(ys[n-1], xs[n])` for $n \geq 1$.

The INT→INT lambdas `(+1)`, `(-1)`, `(*2)`, `(/2)`, `(*(-1))`, `(**2)`, `(*3)`, `(/3)`, `(*4)`, `(/4)` provided by our DSL map integers to integers in a self-explanatory manner. The INT→BOOL lambdas `(>0)`, `(<0)`, `(%2==0)`, `(%2==1)` respectively test positivity, negativity, evenness and oddness of

the input integer value. Finally, the INT→INT→INT lambdas (+), (−), (*), MIN, MAX apply a function to a pair of integers and produce a single integer.

As an example, consider the function SCANL1 MAX, consisting of the higher-order function SCANL1 and the INT→INT→INT lambda MAX. Given an integer array a of length $L$, this function computes the running maximum of the array a. Specifically, it returns an array b of the same length $L$ whose $i$-th element is the maximum of the first $i$ elements in a.

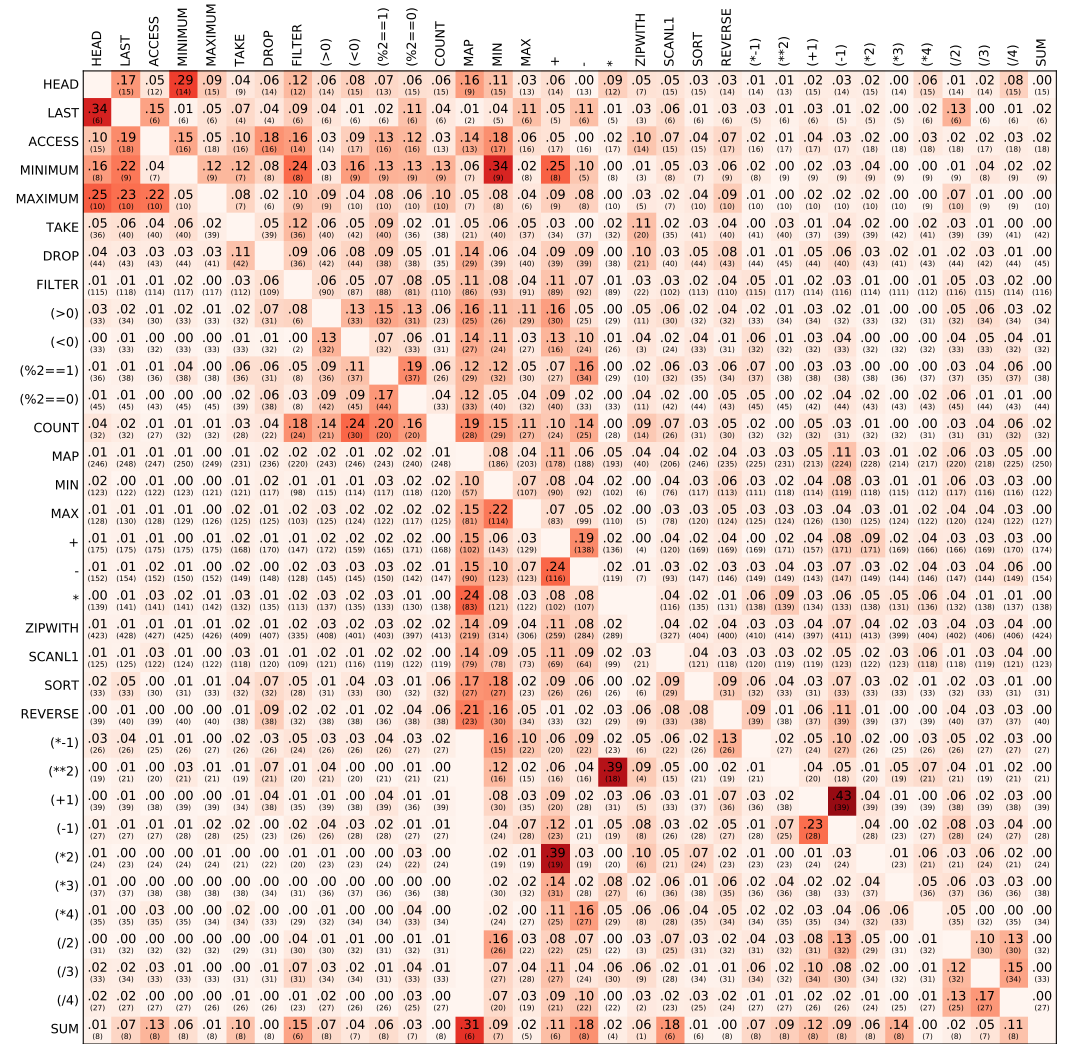

Figure 9: Conditional confusion matrix for the neural network and test set of $P = 500$ programs of length $T = 3$ that were used to obtain the results presented in Table 1. Each cell contains the average false positive probability (in larger font) and the number of test programs from which this average was computed (smaller font, in brackets). The color intensity of each cell's shading coresponds to the magnitude of the average false positive probability.

# G ANALYSIS OF TRAINED NEURAL NETWORKS

We analyzed the performance of trained neural networks by investigating which program instructions tend to get confused by the networks. To this end, we looked at a generalization of confusion matrices to the multilabel classification setting: for each attribute in a ground truth program (rows) measure how likely each other attribute (columns) is predicted as a false positive. More formally, in this matrix the $(i, j)$-entry is the average predicted probability of attribute $j$ among test programs that *do*

possess attribute $i$ and *do not* possess attribute $j$. Intuitively, the $i$-th row of this matrix shows how the presence of attribute $i$ confuses the network into incorrectly predicting each other attribute $j$.

Figure 9 shows this conditional confusion matrix for the neural network and $P = 500$ program test set configuration used to obtain Table 1. We re-ordered the confusion matrix to try to expose block structure in the false positive probabilities, revealing groups of instructions that tend to be difficult to distinguish. Figure 10 show the conditional confusion matrix for the neural network used to obtain the table in Fig. 3a. While the results are somewhat noisy, we observe a few general tendencies:

- There is increased confusion amongst instructions that select out a single element from an array: HEAD, LAST, ACCESS, MINIMUM, MAXIMUM.

- Some common attributes get predicted more often regardless of the ground truth program: FILTER, (>0), (<0), (%2==1), (%2==0), MIN, MAX, (+), (-), ZIPWITH.

- There are some groups of lambdas that are more difficult for the network to distinguish within: (+) vs (-); (+1) vs (-1); (/2) vs (/3) vs (/4).

- When a program uses (**2), the network often thinks it's using (*), presumably because both can lead to large values in the output.

Figure 10: Conditional confusion matrix for the neural network and test set of $P = 500$ programs of length $T = 5$. The presentation is the same as in Figure 9.

