# Peer review of "DeepCoder: Learning to Write Programs"

_ICLR 2017 — accepted_

[Public Comment · Edward Grefenstette · 04 Nov 2016]
**Nice**

Just had a quick scan over this paper. It's very cool to see program induction models that provide interpretable programs rather than a set of weights. Looking forward to reading this in more depth.

Two relevant pieces of work you may wish to read/refer to here (full disclosure: they are from our group so this is a shameless plug):

1) Program generation from text with efficient marginalisation over multiple forms of attention:
Latent Predictor Networks for Code Generation, Ling et al, ACL 2016.

2) Learning pushdown automata and other data structures, similar to the Joulin & Mikolov (NIPS 2015) paper you cite:
Learning to transduce with unbounded memory, Grefenstette et al. (NIPS 2015)

[Public Comment · Surya Bhupatiraju · 07 Dec 2016]
**Great!**

This paper incorporates the PL/ML sides really nicely, and I think it's plausible that highly hybridized approaches such as this might become more and more popular. Great work!

A few questions/comments/things I was wondering:
1. Were you able to experiment with longer programs or with larger integer ranges, or were there fundamental limitations that prevented this? If not, how did this degrade/improve results?
2. C_hat is only computed once per input/output pair and not recomputed as the search produces intermediate programs; was there a reason to not include any partial conditioning, or would this likely not help much?
3. It's true that the augmented searches reach 20% accuracy on the test set much faster than baseline methods, but all methods seem to converge to 100% solving of the test set at the same time -- do you have any conjectures as to why this might be true? It seems like learning slows down enough for the baseline to catchup.

[Public Comment · (anonymous) · 11 Dec 2016]
**Experimental Eval?**

I read the paper in some depth and the idea has some potential, however, the main claim is not substantiated well. Concretely, the 3rd contribution is problematic as the baselines here are very weak and are not what is used in state of the art synthesizers. 

To substantiate point 3, you need to actually compare against a relevant synthesizer. For instance, for the type of programs you use, I would not use sketch, you can select lambda2 (PLDI'15) as a baseline: take the implementation insert your heuristic in there and see what the results would be.

[Official Review · AnonReviewer3 · rating 7 · confidence 2 · 16 Dec 2016]
**An interesting approach. I have several questions that would like the authors to address**

This paper presents an approach to learn to generate programs. Instead of directly trying to generate the program, the authors propose to train a neural net to estimate a fix set of attributes, which then condition a search procedure. This is an interesting approach, which make sense, as building a generative model of programs is a very complex task.

Faster computation times are shown in the experimental section with respect to baselines including DFS, Enumeration, etc. in a setup with very small programs of length up to 5 instructions have to be found. 
It is not clear to me how the proposed approach scales to larger programs, where perhaps many attributes will be on. Is there still an advantage?

The authors use as metric the time to find a single program, whose execution will result in the set of 5 input-output pairs given as input. However, as mentioned in the paper, one is not after a generic program but after the best program, or a rank list of all programs (or top-k programs) that result in a correct execution.
Could the authors show experiments in this setting? would still be useful to have the proposed approach? what would the challenges be in this more realistic scenario?

In the second experiment the authors show results where the length of the program at training time is different than the length at test time. However, the results are shown when only 20% of the programs are finished. Could you show results for finding all programs? 

The paper is missing an analysis of the results. What type of programs are difficult? how often is the NNet wrong? how does this affect speed? what are the failure modes of the proposed method?

The authors proposed to have a fix-length representation of the each input-output pair, and then use average pooling to get the final representation. However, why would average pooling make sense here? would it make more sense to combine the predictions at the decoder, not the encoder?

Learning from only 5 executions seems very difficult to me. For programs so small it might be ok, but going to more difficult and longer programs this setting does not seem reasonable. 

In summary an interesting paper. This paper tackles a problem that is outside my area of expertise so I might have miss something important.

[Official Review · AnonReviewer1 · rating 6 · confidence 4 · 17 Dec 2016]
**My thoughts**

The paper presents a technique to combine deep learning style input-output training with search techniques to match the input of a program to the provided output. Orders of magnitude speedup over non-augmented baselines are presented.

Summary:
———
The proposed search for source code implementations based on a rather small domain specific language (DSL) is compelling but also expected to some degree

Quality: The paper is well written.
Clarity: Some of the derivations and intuitions could be explained in more detail but the main story is well described.
Originality: The suggested idea to speed up search based techniques using neural nets is perfectly plausible.
Significance: The experimental setup is restricted to smaller scales but the illustrated improvements are clearly apparent.

Details:
————
1. The employed test set of 100 programs seems rather small. in addition the authors ensure that the test set programs are semantically disjoint from the training set programs. Could the authors provide additional details about the small size of the test set and how to the disjoint property is enforced?

2. The length of the programs is rather small at this point in time. A more detailed ablation regarding the runtime seems useful. The search based procedure is probably still the computationally most expensive part. Hence the neural net provides some additional prior information rather than tackling the real task.

[Official Review · AnonReviewer2 · rating 6 · confidence 4 · 21 Dec 2016]
**A relatively simple and effective idea for predicting programs from input output pairs**

This is a good paper, well written, that presents a simple but effective approach to predict code properties from input output pairs. 

The experiments show superiority to the baseline, with speedup factors between one to two orders of magnitude. This is a solid gain!

The domain of programs is limited, so there is more work to do in trying such ideas on more difficult tasks. Using neural nets to augment the search is a good starting point and a right approach, instead of generating full complex code.

I see this paper as being above the threshold for acceptance.

[Author Response · Matej Balog · 28 Dec 2016]
**General response**

Thank you to all the reviewers for their time and comments. To summarize, the reviews say this is an interesting and well-written paper. The biggest issue raised in the reviews is that the scale of programs is said to be small (we disagree), and there are questions about how the approach will scale to more complicated programs (this is a big open problem, which--as discussed in Sec 7--we agree we don't solve, but we believe that DeepCoder lays a solid foundation upon which to scale up from.)

We would like to respond to these issues in a bit more detail:

1. DeepCoder is solving problems that are significantly more complex than those tackled in machine learning research (e.g., as can be found in [1-6]), and it can significantly speed up strong methods from the programming languages community. DeepCoder scales up better than any method we are aware of for solving simple programming competition style problems from I/O examples.

2. Inducing large programs from weak supervision (in this case input-output examples) is a major open problem in computer science. We don't claim to solve this problem, and it would be a major breakthrough if we did. There clearly need to be many significant research contributions to get there. DeepCoder develops two ideas that are generally useful: 1. learning bottom-up cues to guide search over program space, and 2. building a system that sits on top of and improves the strongest search-based techniques. The benefit is that as new search techniques are developed, the DeepCoder framework can be added on top, and it will likely provide significant additional improvements (we have now shown large improvements over 3 different search techniques: 2 from the original paper plus the \lambda^2 system requested in reviewer comments).

3. There is a question of whether there is enough information in input-output examples to induce large programs, and we agree that there is probably not enough information using strictly the formulation in this paper (see Discussion). However, it is straightforward to extend DeepCoder: First, by using natural language as an additional input to the encoder, we can learn to extract more information about what the program is expected to do, and how it is expected to do it. Second, generating data from a trained generative model of source code will cause the system to favor generating programs that are likely under the trained model, which will help with the ranking problem. Third, the definition of attributes can be expanded to be richer (e.g., ranking the different instructions per position in the program, or combinations of instructions), to enable the neural network component to solve more of the problem (and thus rely less on the search). Together, we believe these directions represent a clear path towards scaling up program synthesis even further by leveraging the key ideas in DeepCoder.


[1] Graves, A., Wayne, G. and Danihelka, I., 2014. Neural turing machines. arXiv preprint arXiv:1410.5401.

[2] Zaremba, W., Mikolov, T., Joulin, A. and Fergus, R., 2015. Learning simple algorithms from examples. arXiv preprint arXiv:1511.07275.

[3] Andrychowicz, M. and Kurach, K., 2016. Learning Efficient Algorithms with Hierarchical Attentive Memory. arXiv preprint arXiv:1602.03218.

[4] Riedel, S., Bošnjak, M. and Rocktäschel, T., 2016. Programming with a Differentiable Forth Interpreter. arXiv preprint arXiv:1605.06640.

[5] Gong, Q, Tian, Y, Zitnick, C. L., Unsupervised Program Induction with Hierarchical Generative Convolutional Neural Networks. ICLR 2017 Submission.

[6] Kurach, K., Andrychowicz, M. and Sutskever, I., 2015. Neural random-access machines. arXiv preprint arXiv:1511.06392.

[Author Response · Marc Brockschmidt · 16 Jan 2017]
**Summary of changes in 1st and 2nd revision**

We have now uploaded two revisions of our submission, taking the discussion and reviewer feedback here into account. Most notably, we have extended the experimental evaluation to also consider the Lambda2 program synthesis system as back-end solver, and furthermore extended the considered test set of programs to synthesize. We also updated the text in several places to include the clarifications we provided in the comments here, and to cite more related work.

Finally, we are working towards another revision that will include a analysis of failure cases and hope to publish it before the end of the week.

[Final Decision · Program Chairs · 06 Feb 2017]
**ICLR committee final decision**

This is a well written paper that attempts to craft a practical program synthesis approach by training a neural net to predict code attributes and exploit these predicted attributes to efficiently search through DSL constructs (using methods developed in programming languages community). The method is sensible and appears to give consistent speedups over baselines, though its viability for longer programs remains to be seen. There is potential to improve the paper. One of the reviewers would have liked more analysis on what type of programs are difficult and how often the method fails, and how performance depends on training set size etc. The authors should improve the paper based on reviewer comments.